# A Five Year Randomized Controlled Trial to Assess the Efficacy and Antibody Responses to a Commercial and Autogenous Vaccine for the Prevention of Infectious Bovine Keratoconjunctivitis

**DOI:** 10.3390/vaccines10060916

**Published:** 2022-06-09

**Authors:** Matthew M. Hille, Matthew L. Spangler, Michael L. Clawson, Kelly D. Heath, Hiep L. X. Vu, Rachel E. S. Rogers, John Dustin Loy

**Affiliations:** 1School of Veterinary Medicine and Biomedical Sciences, University of Nebraska, Lincoln, NE 68583-0907, USA; mhille@unl.edu (M.M.H.); kheath3@unl.edu (K.D.H.); 2Department of Animal Science, University of Nebraska, Lincoln, NE 68583-0908, USA; mspangler2@unl.edu (M.L.S.); hiepvu@unl.edu (H.L.X.V.); 3United States Meat Animal Research Center, Clay Center, NE 68933, USA; mike.clawson@usda.gov; 4Nebraska Center for Virology, University of Nebraska, Lincoln, NE 68583-0900, USA; 5Department of Statistics, University of Nebraska, Lincoln, NE 68583-0963, USA; rachel.rogers@huskers.unl.edu

**Keywords:** bovine pinkeye, vaccine, *Moraxella bovis*, *Moraxella bovoculi*, *Mycoplasma bovoculi*, infectious bovine keratoconjunctivitis

## Abstract

A randomized control trial was performed over a five-year period to assess the efficacy and antibody response induced by autogenous and commercial vaccine formulations against infectious bovine keratoconjunctivitis (IBK). Calves were randomly assigned each year to one of three arms: an autogenous vaccine treatment that included *Moraxella bovis* (*M. bovis*), *Moraxella bovoculi*, and *Mycoplasma bovoculi* antigens, a commercial *M. bovis* vaccine treatment, or a sham vaccine treatment that consisted only of adjuvant. A total of 1198 calves were enrolled in the study. Calves were administered the respective vaccines approximately 21 days apart, just prior to turnout on summer pastures. Treatment effects were analyzed for IBK incidence, retreatment incidence, 205-day adjusted weaning weights, and antibody response to the type IV pilus protein (pili) of *M. bovis* as measured by a novel indirect enzyme-linked immunosorbent screening assay (ELISA). Calves vaccinated with the autogenous formulation experienced a decreased cumulative incidence of IBK over the entire study compared to those vaccinated with the commercial and sham formulations (24.5% vs. 30.06% vs. 30.3%, respectively, *p* = 0.25), and had less IBK cases that required retreatment compared to the commercial and sham formulations (21.4% vs. 27.9% vs. 34.3%, respectively, *p* = 0.15), but these differences were not significant. The autogenous formulation induced a significantly stronger antibody response than the commercial (*p* = 0.022) and sham formulations (*p* = 0.001), but antibody levels were not significantly correlated with IBK protection (*p* = 0.37).

## 1. Introduction

Infectious bovine keratoconjunctivitis (IBK) is the most commonly diagnosed ocular disease of cattle [1,2]. Cattle producers and veterinarians often refer to the disease simply as “pinkeye”, although this colloquial term can include non-infectious diseases of the eye such as foreign bodies, trauma, or intraocular diseases. The subjectivity of the term pinkeye may be due to a lack of a widely accepted case definition for IBK. Clinical signs for IBK often include epiphora, blepharospasm, conjunctival swelling, and photophobia with more severe cases developing corneal ulceration, buphthalmos, and even rupture of the globe [3,4]. Recently, an epidemiological case definition was proposed for IBK that included; a disease with high morbidity (at least 2% in calves or 0.6% in adult animals) that rapidly disseminates within a herd (average time course of 30 days), with animals displaying clinical signs that are restricted to the eye(s) to include keratitis and/or conjunctivitis, with at least 10% of the lesions displaying corneal ulceration [4]. IBK has a significant economic impact on the cattle industry due to the need for antibiotics, labor required for carrying out treatments, decreased feed efficiency, decreased average daily gain, costs associated with prevention strategies, and animal welfare concerns [1,5,6].

To date, *Moraxella bovis* (*M. bovis*) is the only bacterium with an experimentally demonstrated causal relationship to IBK, with the disease being reproduced after ocular inoculation of cattle [7,8,9]. A closely related Gram negative bacterium, *Moraxella bovoculi* (*M. bovoculi*) is cultured from IBK lesions with a greater frequency than *M. bovis* [10], but thus far has failed to produce IBK or IBK-like disease in an experimental model [11]. Various other bacteria and viruses have also been associated with IBK outbreaks, especially *Mycoplasma bovoculi* and bovine herpesvirus-1. These agents have been shown to be capable of inducing ocular disease under experimental conditions, but with clinical signs that are slightly different than those most commonly associated with IBK [12,13,14].

Utilizing vaccination as a prevention strategy for IBK is relatively common among producers within the United States. There are numerous fully USDA licensed *M. bovis*-based commercial vaccines available, a single conditionally approved *M. bovoculi*-based vaccine (USDA CVM code: 2A77.00, Addison Biological Laboratory, Inc., Fayette, MO, USA. #355), as well as laboratories that offer custom autogenous vaccine formulations that can include a variety of antigens [15]. Studies examining the efficacy of these vaccines in field trials have historically had mixed results, even occasionally exhibiting an increase in IBK incidence among vaccinated animals compared to unvaccinated animals [16,17,18,19,20,21,22]. Organisms that are often associated with IBK lesions, yet remain unproven as causatives agents, suggest the disease may be multifactorial in terms of etiology and this may explain the lack of efficacy seen with vaccines that use a limited number of antigens. Vaccines that employ a large number of antigens within a formulation may be reducing the response magnitude to individual antigens as well. This phenomenon was previously shown in a study that examined antibody titers in animals vaccinated with monovalent or multivalent fimbrial antigens, some of which included fimbriae from a strain of *M. bovis* [23]. In the study, the antibody levels to individual antigens decreased as the number of fimbrial antigens in the vaccine was increased. There may also be other factors such as virulence factors, and environmental or host-associated risk factors that have yet to be discovered. Additionally, recent work in *Moraxella* spp. obtained from bovine eyes has shown substantial genetic diversity within, and between species, as well as evidence of interspecies recombination [24,25]. Taken together, these facts point to a large number of possible etiologies for IBK, and thus a large number of potential vaccine antigen candidates. Both *M. bovis* and *M. bovoculi* express a type IV pilus (pili) protein that enable the bacteria to adhere to the corneal surface [26,27,28,29]. Pili have been shown to play a critical role in the formation of biofilms by *M. bovis* [29]. Given the relatedness of the species, the ocular microenvironment in which they inhabit, and the evidence for pili in *M. bovoculi*, the same pili role is likely true for *M. bovoculi*, although this has not been confirmed experimentally.

A total of seven serogroups have been described with respect to *M. bovis* pili, and protection from disease in a challenge model using pili-based vaccines was shown to be serogroup specific [30,31]. In a separate study, animals vaccinated with a monovalent *M. bovis* vaccine and challenged with isolates from the same serogroup, had isolates recovered from the eyes six days later that expressed pili of a different serogroup, presumably as a result of immunological pressure [32]. Two forms of the *M. bovis* pilus protein, Q (quick) and I (intermediate) have been shown to be the result of gene inversion of the pilus gene [33,34]. The Q form has been shown to be more efficient in corneal attachment, and therefore, more pathogenic. These two forms further increase the variability of antigens possible for *M. bovis*. An exceptionally large repertoire of surface antigens that can be variably expressed poses a potentially severe obstacle to efficacious vaccine development. Too few antigens within a formulation would be problematic if the antigens do not produce sufficient cross reactivity, whereas too many antigens within a formulation risks minimizing the immune response to individual antigens.

The objective of this study was two-fold. First, a randomized controlled trial was designed to assess potential differences in efficacy between the autogenous and commercial vaccine formulations on IBK incidence (primary outcome) as well as treatment effects on IBK treatment success and adjusted weaning weights (secondary outcomes). We also compared the antibody response of calves against the pilus protein of *M. bovis* strain Epp-63 (300). If a particular vaccine formulation were to promote a more robust antibody response to this known *M. bovis* virulence factor, this knowledge would be beneficial in guiding current vaccine strategy use, as well as help shape the formulation of future vaccines.

## 2. Materials and Methods

### 2.1. Animals and Data Collection

The calves used for this study were the annual offspring of a beef teaching herd owned by the University of Nebraska-Lincoln that is housed and managed at the Eastern Nebraska Research, Extension, and Education Center (ENREEC) in Saunders County, NE, USA. The number of calves enrolled in the study (2016–2020) totaled 1198, averaging 240 per year. The breed of calves was composed of approximately 18% purebred Angus, less than 1% purebred Simmental, and 81% crossbred with varying percentages of Angus, Red Angus, Simmental, and Gelbvieh. During the breeding season, the animals were managed in three separate groups including black purebred Angus, black hybrid, and red hybrid management groups. After the bulls were removed in early July, the herds were comingled for the remainder of the grazing period until weaning in early October. The age of calves at the initial vaccination ranged from 10–98 days. The calves were monitored for clinical signs of IBK throughout the grazing period and IBK cases were recorded on the date they were observed. A diagnosis of IBK was made in animals that displayed clinical signs confined to the eye(s) that included; corneal edema, conjunctivitis, epiphora, blepharospasm, corneal ulceration, or corneal neovascularization. Retreatment was defined as cases of IBK that showed no clinical improvement, or whose clinical signs worsened after a post-treatment interval of 72 h. Cases that met this definition were treated with an antibiotic of a different class compared to the initial treatment. Data collected for each calf included calf ID number, date of birth, incidence(s) of IBK, sex, hide color, breed, and weaning weight. Adjusted weaning weights were supplied by the American Simmental Association and were adjusted to a common age of 205-days and for age of dam. Calves that died for any reason prior to weaning were excluded from analysis. Data from years that exhibited a burden of IBK consistent with the epidemiologic case definition proposed by Kneipp [4] were included in the final analysis. Therefore, data from 2016 and 2018 were omitted due to a low incidence of IBK during those years. The total number of calves enrolled in each treatment group for the years analyzed in this study are shown in Appendix A. All calves were treated with a macrolide antibiotic in the summer of 2018 due to an outbreak of pneumonia, which most likely affected the incidence of IBK that year. The number of calves included in the final data analysis from 2017, 2019, and 2020 totaled 672 (56.09% of total initially enrolled). The protocols involved in this study were reviewed and approved by the Institutional Animal Care and Use Committee of the University of Nebraska (#1174).

### 2.2. Vaccine Treatments

In 2016, calves were randomly assigned to one of the three vaccine treatment groups. Eligible calves were examined and ensured to be free of IBK-like lesions prior to study enrollment. The calves received either 2 mL of a commercially available vaccine containing eight strains of *M. bovis* per label instructions (Ocu-guard MB-1, Boehringer Ingelheim Vetmedica, St. Joseph, MO, USA), 3 mL of a custom autogenous vaccine containing *M. bovis*, *M. bovoculi*, and *Mycoplasma bovoculi* antigens with a proprietary oil-in-water adjuvant emulsion per label instructions (Phibro Animal Health Corporation, Teaneck, NJ, USA), or 3 mL of a sham vaccine containing only the oil-in-water adjuvant (Emulsigen-D, Phibro Animal Health Corporation). Isolates included in the annual autogenous vaccine formulation were chosen to capture maximum diversity based on surface protein electrophoresis conducted by the vaccine manufacturer and biotyping results using matrix-assisted laser desorption/ionization mass spectrometry (MALDI-TOF MS) profiles performed at the Nebraska Veterinary Diagnostic Center. The utility of MALDI-TOF MS biotyping based on a mean spectrum profile (MSP) relatedness for epidemiological typing has been previously described [35]. In subsequent years, calves received the same vaccine treatment as their dam. Animals were processed individually in a hydraulic squeeze chute for vaccine administration. Calves received an initial dose of their respective vaccine treatment, followed by a second dose approximately 21 days later. The timing of the second dose varied slightly (±5 days) depending upon staff availability and weather conditions. Blinding of the cattle processing staff was achieved by a third party transferring the vaccines to bottles without identifying information that were labeled either treatment 1, treatment 2 or treatment 3, which was randomized each year.

### 2.3. Blood Collection

Paired serum samples were obtained approximately 4–5 weeks apart from a subset of 20 calves from each vaccine treatment group, selected randomly, each year. The first serum sample was obtained immediately preceding the first vaccine dose and the second serum sample was obtained 2–3 weeks after the second dose. Each animal’s head was restrained using a halter and blood samples were obtained from the jugular vein. The blood was allowed to clot, then centrifuged for harvesting of the serum which was then stored at −80 °C until testing.

### 2.4. Enzyme Linked Immunosorbent Screening Assays

An indirect enzyme-linked immunosorbent screening assay (ELISA) was developed to screen serum from a subset of calves (*n* = 15/treatment/year) for the presence of IgG specific for the Q form of the type IV pilus protein of *M. bovis* strain Epp-63 (300) (Epp63) [36]. A custom full-length recombinant protein (rPilA) (GenScript, Piscataway, NJ, USA), with a total length of 157 amino acids was used as target antigen for the rPilA ELISA. A 38 amino acid section of the pilus sequence was randomly scrambled, and a synthetic peptide with the scrambled sequence (GenScript, Piscataway, NJ, USA) was used as negative control antigen. The sequences of each ELISA antigen are shown in Appendix A.

Immulon 2 HB 96-well flat-bottom plates (Thermo Fisher Scientific, Waltham, MA, USA) were coated with 100 µL of 10 µg/mL (1 µg) rPilA antigen in phosphate buffered saline (Thermo Fisher, Waltham, MA, USA) with 0.05% *w*/*v* NaN_3_ (Sigma Aldrich, St. Louis, MO, USA) and incubated overnight at 4 °C. Remaining binding sites within the wells were blocked with 135 µL of ChonBlock™ ELISA buffer (Chondrex Inc., Woodinville, WA, USA) for one hour at room temperature. Serum samples diluted 1:400 in ChonBlock™ ELISA buffer (Chondrex Inc., Woodinville, WA, USA) to a final volume of 100 µL, were incubated for two hours at room temperature. Secondary antibody incubation involved 100 µL of a 1:5000 dilution of peroxidase-conjugated rabbit anti-bovine IgG (H+L) (Jackson Immunoresearch, West Grove, PA, USA) in ChonBlock™ detection antibody buffer (Chondrex, Inc., Woodinville, WA, USA) incubated for 1 h at room temperature. A 100 µL volume of 3,3′,5,5′-tetramethylbenzidine substrate (Seracare Life Sciences, Milford, MA, USA) was introduced to each well and the plate was placed on a plate rocker for 10 min at room temperature. After ten minutes, the peroxidase reaction was stopped by the addition of 100 µL of 2 M H_2_SO_4_ (Sigma Aldrich, St. Louis, MO, USA). Light absorbance results, as measured by optical density (OD) were obtained using an Epoch2 microplate reader (BioTek, Winooski, VT, USA) with a measurement wavelength setting of 450 nm. Results were normalized by subtracting the OD signal to the scrambled peptide from the OD signal to the rPilA. Each assay had duplicates for both the rPilA and scrambled (negative control) antigen for each serum sample tested. Plates were washed three times with Tris-buffered saline solution containing 0.05% *v*/*v* tween (Fisher Scientific, Hampton, NH, USA) between the antigen, blocking, primary antibody (serum), and secondary antibody steps. Plates were washed five times after the secondary antibody incubation step. A negative control primary antibody sample of a 1:400 dilution of fetal bovine serum was included in each ELISA assay.

### 2.5. Statistical Analysis

Analysis of treatment effect on IBK incidence was performed using SAS and a logistic regression model with a logit link function where the resulting predicted probability was the probability of IBK of the calf. The logit link function then related the linear predictor to the predicted probability. The Tukey–Kramer adjustment was used when conducting pairwise comparisons, thus the *p* values for comparison of years and management groups are an adjusted *p* value. We chose to fit the binomial model for IBK in this study as it allowed us to examine all variables such as sex, hide color, and management group for effect, and not just vaccine treatment. These variables were included as fixed blocks, in order to account for variation when evaluating vaccine treatment. The model used in this study is as follows:



(1)
$$y_{ijklm}\sim Binomial(p_{ijklm})\;\eta_{ijklm}=\log\left(\frac{p_{ijklm}}{1-p_{ijklm}}\right)=\eta +\tau_{i}+h_{j}+s_{k}+b{\left(h\right)}_{jl}+t_{m}$$



Symbols are defined as follows: η is the overall intercept, $\tau_{i}$ is the effect of the ith treatment, $h_{j}$ is the effect of the jth hide color, $s_{k}$ is the effect of the kth sex,$\;b{\left(h\right)}_{jl}$ is the effect of the $l^{th}$ management group nested with hide color, and $t_{m}$ is the effect of the mth year. Year, treatment, hide color, sex, and breed were treated as fixed. For analysis on treatment effect on retreatment rates, the binomial model was used and only calves that developed IBK were analyzed. Relative risk of IBK by vaccine treatment was analyzed using the formula previously described [37], and is as follows:(2)$$Relative\;Risk=\frac{1-P_{2}}{1-P_{1}}\times Odds\;Ratio$$

ELISA data used a multivariate analysis without binary transformation. The ELISA OD value was the response and the ELISA assay was a predictor. The following model was used: ${\mathrm{ELISA}}_{ijklmnp}=e_{n}+\tau_{i}+h_{j}+s_{k}+b{\left(h\right)}_{jl}+t_{m}+e\tau_{ni}+\epsilon_{ijklmnp}$. Where $e\tau_{ni}$ is the interaction between the ELISA and the treatment, and ϵ~N(0,Σ), where Σ is an unstructured covariance matrix. All other variables were as defined above. A linear model was used to examine the effect of treatment on 205-day adjusted weaning weight. The model used was:$\;w_{ijklmp}=\tau_{i}+h_{j}+s_{k}+b{\left(h\right)}_{jl}+t_{m}+\epsilon_{ijklmp}$, with symbol abbreviations as defined above. Here, $\epsilon_{ijklmp}$ is the residual term that assumed to be distributed $N\left(0,\sigma^{2}\right)$.

To determine the utility of rPilA ELISA OD results as a predictor of IBK, the binomial model as above was used, with the inclusion of rPilA ELISA. The resulting equation was as follows: $\eta_{i}=\log\left(\frac{p_{i}}{1-p_{i}}\right)=-1.275+1.0236*x_{i}=\eta +\beta_{e}x_{i}$. Leading to the probability of IBK being defined as: $P\left(\mathrm{IBK}\right)=\frac{1}{1+e^{-\left[-1.275+1.0236*x\right]}}=\frac{1}{1+e^{-\left[\eta +\beta_{e}x\right]}}$.

## 3. Results

### 3.1. IBK Incidence

IBK incidence data from 2016 and 2018 were omitted from analysis after these years did not meet the epidemiological IBK case definition due to low incidence of disease. Over the three years of data analyzed (2017, 2019, and 2020), the majority of cases were diagnosed in the peak summer months of July, August, and September with some variation between years (Figure 1).

The cumulative incidence for IBK cases over the entire study, regardless of vaccine, was 28.42% with 191 cases of IBK being diagnosed among the 672 calves. The yearly incidence ranged from 18.4% to 43.55% with 2020 having more than twice the number of IBK cases compared to either 2017 or 2019 (Figure 2).

Calves that received the autogenous vaccine formulation tended to have a lower incidence of IBK compared to the commercial and sham groups (24.5% vs. 30.6% vs. 30.3%, respectively), but this difference was not statistically significant (Figure 3). Bull calves had a higher incidence of IBK (30.4%) vs. heifers (26.3%), but the difference was not significant (*p* = 0.21, data not shown). The estimate for the black hybrid management group was significantly higher than the black purebred Angus group (*p* = 0.02, data not shown). The estimate for the red hybrid group was intermediate, and not statistically different from either black purebred Angus or the black hybrid management groups.

Mean probability estimates of IBK development for all treatment group and phenotype combinations in the study are summarized in Appendix A. Relative risk of IBK among vaccine treatment comparisons when averaging over treatment, sex, year, and management group nested within hide color, is summarized in Figure 4.

### 3.2. Retreatment Rate

Of the calves that developed IBK during the study, we examined the number of calves that required retreatment, and whether the need for retreatment was different among the vaccine treatment groups. No calves needed more than two treatments in this study. The requirement for retreatment was less in the autogenous group (21.4%) vs. the commercial (27.9%) and the sham group (34.4%) but the differences were not significant (*p* = 0.15). Sex did not have a significant effect on retreatment (*p* = 0.07, data not shown). In agreement with overall IBK incidence data, the retreatment estimate was significantly higher in the black hybrid management group compared to the black purebred Angus management group (Adjusted *p* = 0.0477 when comparing the black hybrid and black purebred Angus management groups, data not shown) while the retreatment estimate for the red hybrid management group was not statistically different from either of the black-hided management groups. Over the three-year period analyzed, there were 54 IBK cases that required retreatment, with only two of those cases in the black purebred Angus management group. Mean probability estimates for retreatment of all vaccine treatment group and phenotype combinations are summarized in Appendix A.

### 3.3. Pili Antibody Response

Serum collected after the second vaccination was used to assess the antibody response to a full-length recombinant PilA protein (rPilA). Autogenous vaccines had significantly higher average post-vaccination antibody levels as the ELISA OD was higher (estimate 0.3710) compared to both the commercial and sham groups (estimates 0.2579 and 0.2176, respectively), which were not statistically different (Figure 5). When age at weaning, age at weaning*ELISA, age at weaning*Treatment, and age at weaning*Treatment*ELISA values were included in the equation above, age had an effect on the rPilA ELISA antibody levels (Appendix A). When comparing treatment differences for different age values, the difference between the autogenous and sham group rPilA ELISA values increased until becoming significantly different at 220 days of age at weaning (*p* = 0.007, data not shown).

The rPilA ELISA was used to assess the change in magnitude of antibody levels after vaccination. The change in pre-vaccination to post-vaccination antibody levels to the rPilA antigen varied widely among the calves, with some from each treatment group displaying a decrease in antibody from pre-vaccination to post-vaccination (data not shown). The interassay variability of OD signal using the rPilA ELISA had an average standard deviation of 13.92% when the assay was performed on different days using identical sera (data not shown). Therefore, we chose a 14% increase in OD result as a threshold to characterize an increase in antibody levels. Using this cutoff, the autogenous vaccine had more calves elicit a response (81.33%) than both the commercial and sham vaccines (62.7% and 60.3%, respectively).

### 3.4. Pili Antibody Response and IBK Incidence

Next, we assessed whether the rPilA antibody levels were correlated to IBK protection. Animals that were diagnosed with IBK during the study tended to have lower antibody levels (OD estimate: 0.248, SD: 0.20)) compared to animals that did not develop IBK (OD estimate: 0.294, SD: 0.22), although not to a significant degree (*p* = 0.37).

The logistic regression model equation was used and averaged over treatment, year, sex, and management group nested within breed, when considering the rPilA ELISA OD result in calculating the probability of IBK to determine the utility of the rPilA ELISA on IBK susceptibility prediction. The resulting *p* of 0.3661 indicates the rPilA ELISA result is a poor predictor of IBK susceptibility (data not shown).

### 3.5. Calf Performance

The 205-day adjusted weaning weights were not significantly affected by the presence or absence of IBK diagnosis (*p* = 0.33). The autogenous and sham groups had decreased adjusted weaning weight estimates in calves with IBK (243.88 kg vs. 239.70 kg and 244.03 kg vs. 236.70 kg, respectively), while the commercial group had an increased average weaning weight in calves with IBK (242.58 kg vs. 247.36 kg). Retreatment rate effect on 205-day adjusted weaning weights was approaching significance (*p* = 0.08). The autogenous and commercial groups had decreased adjusted weaning weight estimates in calves that required retreatment (240.86 kg vs. 227.88 kg and 250.66 kg vs. 231.81 kg, respectively), while the sham group had an increased average adjusted weaning weight in calves that required retreatment (233.96 kg vs. 238.43 kg). The 205-day adjusted weaning weight data are summarized in Appendix A. As expected, sex of the calf had a highly significant effect on 205-day adjusted weaning weight (data not shown).

## 4. Discussion

This study is novel as we were able to prospectively follow a herd for five years that historically had issues with IBK, with three of those years meeting the criteria for disease burden. The study used two vaccine treatment arms, thus allowing us to simultaneously compare the two distinct vaccine formulations to a placebo under field conditions. Additionally, there were *Mycoplasma bovoculi* antigens incorporated into the autogenous vaccine formulation, and pili-specific antibody responses induced by each vaccine treatment were examined. While calves vaccinated with the autogenous formulation experienced a decreased incidence of IBK over the entire study compared to those vaccinated with commercial and sham formulations (24.5% vs. 30.06% vs. 30.3%, respectively), and the autogenous formulation had less IBK cases that required retreatment compared to the commercial and sham formulations (21.4% vs. 27.9% vs. 34.3%, respectively), these results did not meet the threshold for statistical significance.

The herd experienced peak IBK incidence in the middle of summer and had a higher cumulative incidence of IBK among bulls, both characteristics that are consistent with previous epidemiological studies of IBK [2,38,39,40]. The effect of management group on IBK incidence and retreatment rate was substantial; however, it is not clear what impact the precise pasture location and/or any genetic characteristics within the management groups may have had in this result. A more detailed analysis incorporating the degree of heterozygosity, fitting breed fractions to better isolate breed differences, and accounting for differences among sires or the inherent differences among the calves themselves would be required to examine any genetic influence on IBK susceptibility.

Our hypothesis was that the vaccine treatments would reduce cumulative IBK incidence over sham vaccination. However, no significant differences were observed for treatment effect on cumulative IBK incidence, but there did appear to be a potential association between the incidence of IBK and the antibody response to the *M. bovis* pili. These observations may help guide future vaccine formulations and experimental studies for IBK under field conditions. The autogenous arm had the lowest cumulative incidence of IBK of all treatment groups. Moreover, the antibody levels to the rPilA antigen, as measured by ELISA, were significantly higher in calves vaccinated with the autogenous formulation, and animals free of IBK tended to have higher antibody levels although this correlation was also not significant. The antibody levels did not differ between the management groups (*p* = 0.63, data not shown), despite the significant differences in IBK incidence, which suggests that additional factors besides just the humoral response to pili account for the IBK incidence differences and may warrant further study. Since the antibody levels did not significantly correlate with IBK protection, it was not surprising that attempts to predict IBK probability using the rPilA ELISA results were not accurate enough to be of any practical clinical value. Several factors may have contributed to this observation including 1; the type IV pilus protein is a poor protective antigen for *M. bovis* and other surface antigens may be more protective or 2; the antibody response specific for the type IV pilus was below a protective threshold, or 3; *M. bovis* strains in diseased eyes expressed a different variant of pili or 4; *M. bovis* was not a significant pathogen in IBK cases during this study. Previous studies have shown, at least under experimental challenge conditions, that a humoral response to the *M. bovis* pili can be protective against experimental *M. bovis* inoculation [31]. This result would suggest that the first explanation above is unlikely to be the case. The discrepancy in our results compared to previous studies may also be due to the field conditions under which the study was performed. The relative abundance of strains that possess the different pili variants in situ in the field is not known. If these variants are regionally specific, or if they are disproportionately represented among virulent *M. bovis* isolates, then a vaccine incorporating strains with specific pili variants would warrant investigation for potential protection given our demonstration in this study that they are indeed antigenic.

It is possible that the non-responder calves in this study may have harbored a higher bacterial load on their ocular surface due to a lack of appreciable antibody. If so, these non-responder calves may serve as a reservoir of bacteria that are capable of negating the antibody response as observed in the remaining calves that did respond. A study that is capable of determining antibody status after vaccination, and the subsequent grouping of calves as responders or non-responders in separate locations for the remainder of the grazing period could test this hypothesis. However, the fact that 60.3% of the negative control group had an increase in antibody levels after being injected with only adjuvant, raises the question as to whether or not the vaccine antigens are the driving force responsible for the changes in antibody level. It is possible that all, or some, of the increase in antibody levels are due to natural exposure at the ocular surface, and not the vaccines themselves. However, the higher rPilA ELISA results in post-vaccination serum in both the autogenous and commercial vaccines compared to the sham group would suggest that the vaccines are responsible for at least a portion of the resulting antibody response.

### Study Limitations

There are several potential study design limitations worth noting that may have had an effect on the results. Firstly, only two vaccine formulations with antigens were examined in this study. It is possible other commercial or autogenous formulations may have performed differently in terms of IBK protection, but increasing the number of formulations would have negatively affected the statistical power of this study given the finite number of animals enrolled per year. Given that all IBK vaccine formulations available currently are whole-cell bacterins, differing only in strain compilation and adjuvant, and the historically poor efficacy of vaccines under experimental field conditions, this would suggest that using different or more vaccine formulations would have had minimal effect on results. Furthermore, we chose to use adjuvant only for the sham treatment in this study instead of a saline control. The adjuvant only formulation was chosen to focus on the impact of antigen makeup between vaccine treatment groups, and not the effect of adjuvant. It is possible that a saline control group may have had significantly different IBK incidence compared to what was observed in the adjuvant only treatment group. Finally, this study utilized a single cow/calf herd, in a specific geographic environment, under a specific management program. Therefore, the effects of geographic impact or management strategies were not a focus of this study and the same study performed in a different location, or under a different management strategy may have yielded different results.

One unexpected result in the analysis for this study was the fact that IBK diagnosis, regardless of vaccine treatment, did not have an effect on weaning weights. While similar results have been reported previously [41], the majority of studies show a detrimental effect of IBK on weight gain [2,5,20]. One potential explanation for our results is that the average severity of IBK cases diagnosis by the ENREEC staff in the current study was less compared to other studies. The ENREEC farm is well staffed, with knowledgeable management, and the calves are visually examined with greater frequency compared to the vast majority of cow/calf herds due to the number of employees available to survey the animals. It is possible that the staff at ENREEC were diagnosing these cases at an earlier disease stage, and potentially minimizing any impact of IBK by treating calves earlier in the time course of disease. Additionally, the current study used age-adjusted weaning weight. It is possible that previous studies that used actual weaning weights could have inadvertently conflated impacts of IBK on weight gain with age given younger calves tend to be lighter when weaning occurs on a common day. In cases of retreatment, the weaning weight was affected by IBK diagnosis in both of the vaccinated groups, which would support the hypothesis of earlier diagnoses for first treatments, since cases of retreatment presumably represent the most severe of cases. These more severe cases may be more analogous to cases of IBK on other farms, where the cases may be diagnosed later in the disease process.

## 5. Conclusions

The current study showed decreased IBK incidence and need for retreatment in the autogenous group compared to the commercial and sham group, but the differences were not significant in either case. These results are consistent with previous studies that examined similar formulations separately, and found no significant benefit of protection. The comparison of antibody levels induced by different vaccine formulations using a novel recombinant protein ELISA assay is a novel aspect of this study. Calves vaccinated with the autogenous formulation had significantly higher IgG antibody levels to the *M. bovis* rPilA antigen than calves that were vaccinated with either the commercial or sham formulation. While we showed positive antigenicity of the *M. bovis* pili, there was no significant correlation between pili-specific antibody levels and IBK protection. However, since IBK cases had lower average antibody levels, humoral immunity may still play a role in terms of IBK protection.

## Figures and Tables

**Figure 1 vaccines-10-00916-f001:**
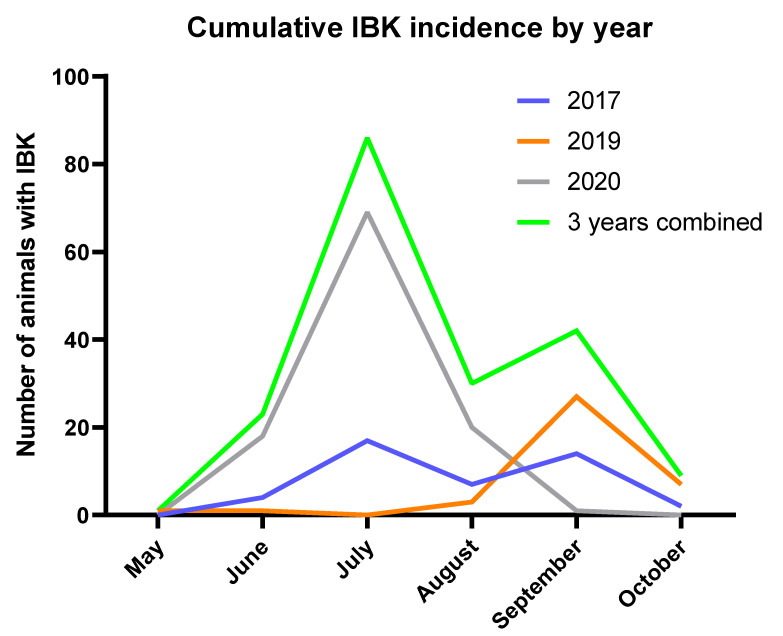
Cumulative IBK incidence by month for years 2017, 2019, and 2020.

**Figure 2 vaccines-10-00916-f002:**
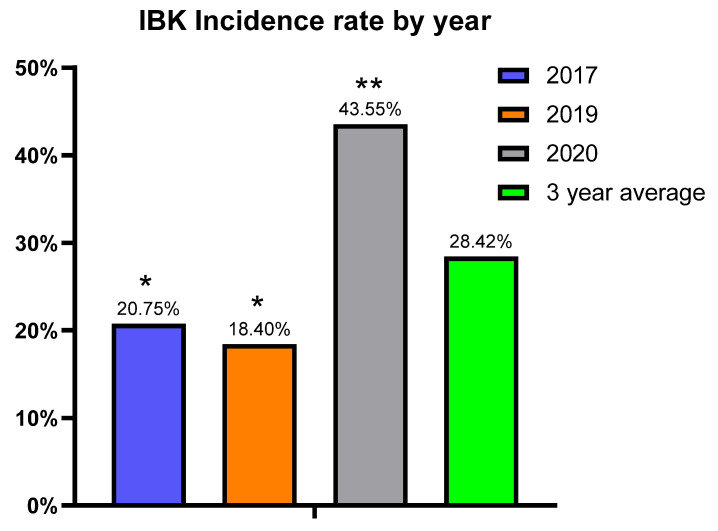
Cumulative incidence of IBK for all calves regardless of vaccine treatment group. Percentages calculated as the number of IBK diagnoses per calves enrolled each year. (*p* ≤ 0.001). * vs. ** indicates significantly different values.

**Figure 3 vaccines-10-00916-f003:**
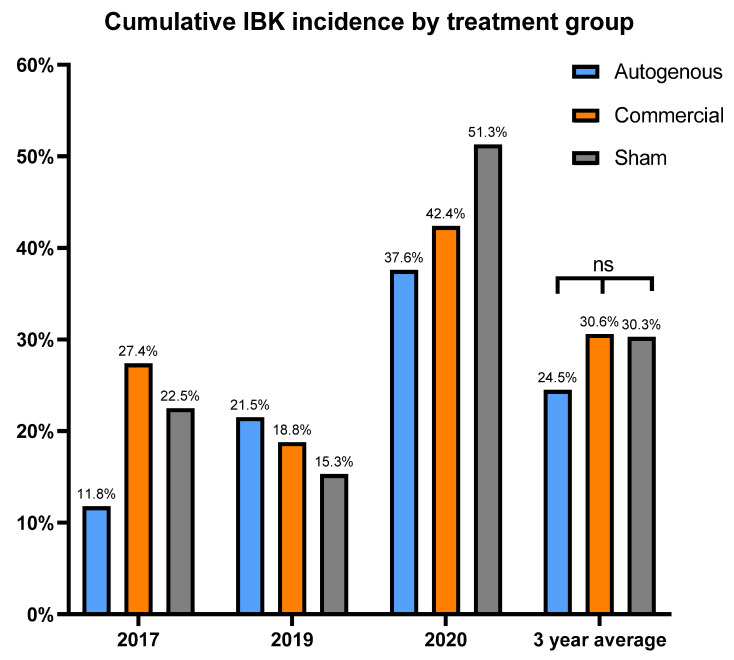
Annual and 3-year average incidence of IBK diagnosis by vaccine treatment group. Percentages calculated as the number of IBK diagnoses per calves enrolled each year. Vaccine treatment did not significantly affect IBK incidence. ns = not significant (*p* = 0.25).

**Figure 4 vaccines-10-00916-f004:**
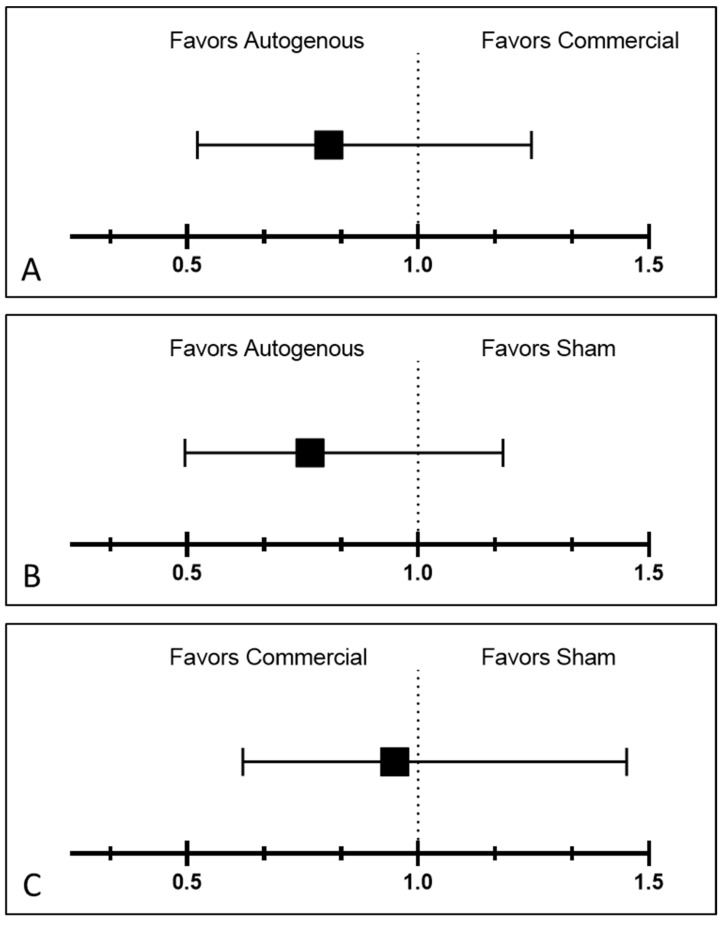
Risk ratio for the effect of treatment on incidence of IBK for the vaccine comparisons in this study. 4 (**A**): Autogenous vs. Commercial: RR = 0.8066 (95% CI 0.5223–1.2455). 4 (**B**): Autogenous vs. Sham: RR = 0.7659 (95% CI 0.4953–1.1842). 4 (**C**): Commercial vs. Sham: RR = 0.9495 (95% CI 0.6208–1.4521). The vertical dotted line at a RR of 1.0 on the X axis correlates to no treatment effect.

**Figure 5 vaccines-10-00916-f005:**
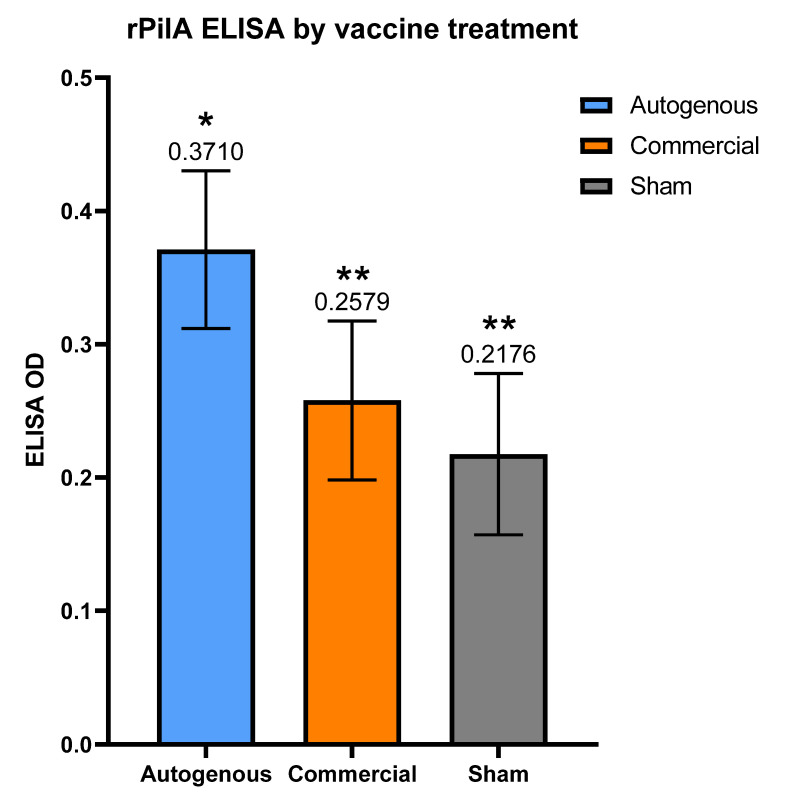
ELISA OD readings by vaccine treatment groups. Adjusted *p* of 0.0224 when comparing Autogenous and Commercial. Adjusted *p* of 0.0013 when comparing Autogenous and Sham. Error bars indicate 95% confidence limits. * vs. ** indicates significantly different values.

## Data Availability

The data presented in this study are available in the Appendix A and upon request from the corresponding author.

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
