# Peer review of "A Five Year Randomized Controlled Trial to Assess the Efficacy and Antibody Responses to a Commercial and Autogenous Vaccine for the Prevention of Infectious Bovine Keratoconjunctivitis"

_vaccines, 2022, doi:10.3390/vaccines10060916_

Round 1

Reviewer 1 Report

The authors conducted a series of experiments to assess the efficacy and antibody responses to a commercial and autogenous vaccine for the prevention of IBK, which has a significant impact on the cattle industry. The authors did a lot of work, however, the manuscript is needs to be carefully revised to improve its clarity and the data presentation.

  1. In 2.1 and 2.2, the author should better to condense these paragraphs into concise sentences, and delete redundant descriptions.
  2. All the figures in this manuscript should better to be drawn with GraphPad or Origin, rather than Excel.

Author Response

The authors conducted a series of experiments to assess the efficacy and antibody responses to a commercial and autogenous vaccine for the prevention of IBK, which has a significant impact on the cattle industry. The authors did a lot of work, however, the manuscript is needs to be carefully revised to improve its clarity and the data presentation.

In 2.1 and 2.2, the author should better to condense these paragraphs into concise sentences, and delete redundant descriptions.

Thank you, these paragraphs have been edited and condensed to eliminate extra wording and/or redundant statements.

2. All the figures in this manuscript should better to be drawn with GraphPad or Origin, rather than Excel.

Thank you, all figures have been redrawn using Graphpad.

Please find an attached, changes tracked and edited version that includes changes described above.  

Reviewer 2 Report

In this study, the authors undertook to compare different vaccines against pinkeye disease in calves during a period of three years.

They compared their own autogenous formulation to a commercially available vaccine and further compared it with the adjuvants of their own make as negative control.

Both vaccines were unsuccessful to prevent disease significantly over background measurements, whereby, their own autogenous vaccine tendential was more effective.

Here, one could actually stop with the manuscript as negative results have many possible and impossible explanations. However, I see value in their carefully researched findings as it became clear to me that current research scientists believe vaccinations against acute infections have been settled. Their hypothesis is, if the vaccine fails then it is the pathogens surface variability which led to the immune surveillance escape. I think, research in general has overlooked the strength of the polyclonal immune responses. This polyclonal response is formed against an enormous variety of overlapping T-cell and humoral epitopes of the corresponding pathogen and a few aas exchanges will not disturb a normal immune response leaving the pathogen unopposed.

Also in this manuscript the MALT associated, specific immune responses are not on the radar of these authors. For the eye as well as the respiratory tract it is not the cytotoxic T-cell responses nor the IgG responses that will prevent the proliferation of the pathogens activated by previous vaccinations. Even though the authors can measure elevated IgG, it is the relatively unspecific IgA responses that can prevent the adherence and the proliferation of the bacteria to the cornea.

I understand that multiple factors play into the development of IBK. However, it seems from this study that macrolide antibiotics prevented IBK cases in 2018. Thus, their pinkeye disease was caused mostly by bacterial infection and therefore, vaccination should have prevented significant disease outcome. However, this was not the case. Unfortunately, the authors do not know the IgA antibody concentrations. It will probably be a major task on its own for bovine research. This appears to me a parallel of a recent publication in Cell where scientists analyzed IgA levels after vaccination against an acute respiratory infect (https://www.ncbi.nlm.nih.gov/pmc/articles/PMC8786601/). Also these scientists were not able to measure elevated IgA against the respiratory tract infection, even though they discovered massive responses of IgG antibodies in the blood after vaccination. And the efficiency of these vaccinations are under discussion in scientific circles. In this line of thought it is also not surprising that for more than 60 years no vaccine could be developed against RSV, also a respiratory tract infect.

Minor adjustments to the manuscript:

To indicate in M&M the numbers of animals added to each of the three groups per year.

Please, also clarify your hypothesis in lines 416-419. As a side note: If a pathogen is overwhelming the immune response, this normally means “game over”.

Author Response

In this study, the authors undertook to compare different vaccines against pinkeye disease in calves during a period of three years.

They compared their own autogenous formulation to a commercially available vaccine and further compared it with the adjuvants of their own make as negative control.

The author’s appreciate the reviewer’s comments.  For clarification,  the autogenous formulation and adjuvant (i.e. sham) used in this study was made by the licensed manufacturer. We provided bacterial strains isolated from the study herd and assisted in strain selection for inclusion, but the company manufactured the vaccine. We purposefully did this to mirror currently available biologics products to what are currently available to producers who utilize USDA licensed autogenous vaccines.

Both vaccines were unsuccessful to prevent disease significantly over background measurements, whereby, their own autogenous vaccine tendential was more effective.

Here, one could actually stop with the manuscript as negative results have many possible and impossible explanations. However, I see value in their carefully researched findings as it became clear to me that current research scientists believe vaccinations against acute infections have been settled. Their hypothesis is, if the vaccine fails then it is the pathogens surface variability which led to the immune surveillance escape. I think, research in general has overlooked the strength of the polyclonal immune responses. This polyclonal response is formed against an enormous variety of overlapping T-cell and humoral epitopes of the corresponding pathogen and a few aas exchanges will not disturb a normal immune response leaving the pathogen unopposed.

Also in this manuscript the MALT associated, specific immune responses are not on the radar of these authors. For the eye as well as the respiratory tract it is not the cytotoxic T-cell responses nor the IgG responses that will prevent the proliferation of the pathogens activated by previous vaccinations. Even though the authors can measure elevated IgG, it is the relatively unspecific IgA responses that can prevent the adherence and the proliferation of the bacteria to the cornea.

We appreciate the reviewer’s comments in regards to efficacy.  We think that even though the results are dispositive, these vaccines are licensed and used widely by producers to mitigate the impact of this disease.  Therefore, the overall findings are beneficial to those who are looking at incorporating these vaccines into their health programs.  We also agree with Reviewer #2’s statements here regarding the immune response as it pertains to IBK. There remain large knowledge gaps on the importance of IgA vs IgG in IBK prevention. Zbrun et al. (DOI: 10.1016/j.rvsc.2011.05.008 ) found no differences in IBK incidence in calves that had significantly higher anti-Moraxella IgA antibody levels. Other studies (including this one) have shown an increase in IgG does not necessarily correlate with disease or protection. One hypothesis is that since Moraxella spp. are normal flora and opportunistic pathogens, that being able to prevent adherence by IgA may be impractical. The levels of IgG are increased substantially once the mucosal barrier is broken, therefore it is possible that a robust IgG response may help reduce disease severity although studies that have looked at this under experimental conditions show mixed results. Angelos (DOI: 10.1016/j.cvfa.2021.03.002 ) provides a good summary of many of these studies, some of which we have cited in our manuscript.

Given that immunological protection is polyclonal, our hypothesis was that the host generated an IgG response to the type IV pilus protein.   This protein is a known virulence factor and previous work had shown associations with disease responses to it.  It is possible that increases in IgG do not correlate well with IgA of the same epitope specificity, and we agree with Reviewer #2’s suspected potential importance of IgA.  We have discussed using the current ELISA framework to try and development IgA results for future studies that are more focused on examining the correlation of protection.  For the current study, our main question was on relative response and how the vaccines affected it.

I understand that multiple factors play into the development of IBK. However, it seems from this study that macrolide antibiotics prevented IBK cases in 2018. Thus, their pinkeye disease was caused mostly by bacterial infection and therefore, vaccination should have prevented significant disease outcome. However, this was not the case. Unfortunately, the authors do not know the IgA antibody concentrations. It will probably be a major task on its own for bovine research. This appears to me a parallel of a recent publication in Cell where scientists analyzed IgA levels after vaccination against an acute respiratory infect (https://www.ncbi.nlm.nih.gov/pmc/articles/PMC8786601/). Also these scientists were not able to measure elevated IgA against the respiratory tract infection, even though they discovered massive responses of IgG antibodies in the blood after vaccination. And the efficiency of these vaccinations are under discussion in scientific circles. In this line of thought it is also not surprising that for more than 60 years no vaccine could be developed against RSV, also a respiratory tract infect.

 We agree that it is likely that the primary etiologic agent is a bacterium, and experimental research supports this finding using gnotobiotic animals.   However, this does not necessarily conclude that vaccination should prevent a significant disease outcome.  Multiple bacterial species, covering much of the bacteria that have had an association with disease in the literature were included in the formulations, and there were non-significant differences observed amongst treatments.  We agree that opportunistic bacterial pathogens and infections on mucosal surfaces, especially in disease complexes such as respiratory and ocular diseases are likely to be challenging targets for effective vaccinations.  One reason for this work was to demonstrate that new tools for prevention of these diseases are needed as currently available solutions may not be adequate, and this is study continues to support that existing vaccines are inadequate to significantly prevent disease.  

Minor adjustments to the manuscript:

To indicate in M&M the numbers of animals added to each of the three groups per year.

A supplementary table (S1) has been added to the submission to show the precise number of calves in each treatment group for each of the years analyzed in this study.

Please, also clarify your hypothesis in lines 416-419. As a side note: If a pathogen is overwhelming the immune response, this normally means “game over”

The first few sentences of this paragraph (Paragraph #4 of Discussion) have been rewritten to try and clarify this hypothesis.

Please see the attached edited and changes tracked revised manuscript which includes these specific changes.  

Reviewer 3 Report

Authors planned a randomized controlled trail to investigate the comparison of two vaccines using different antigens from different pathogens and assessed their efficacy in terms of clinical outcomes of IBK. The trail was conducted for consecutive 5 years while data of only three years are presented due to low IBK incidence and/or outbreaks of pneumonia. The trail is well planned and objectives are very clear.

However, some minor changes suggested are as

Title may be suggestive as: A comparative study to assess the efficacy of an autogenous vaccine with a commercial vaccine against infectious bovine keratoconjunctivitis-A randomized controlled clinical trial.

Line 19. ………administered the respective vaccines approximately 21 days apart……..

Line 25,26. Kindly add p-values in the parentheses while describing the hard data e.g.,  (24.5% vs 30.06% vs 30.3% respectively: P?).

Line26,27. Same may be for Ig response and correlations with IBK.

Line45. The IBK has significant economic impact…..

Line 54-58. Various bacteria and viruses have also been found to be associated with IBK outbreaks, especially Mycoplasma bovoculi and bovine herpesvirus-1 that are  inducing ocular disease, but with clinical signs slightly different from those most commonly associated with IBK (12-14).

Line 101-105……to investigate the potential differences in efficacy between autogenous vaccine and a commercial vaccine in calves on IBK incidence (primary outcome) and IBK treatment success and adjusted weaning weight as secondary outcomes.

Line 106-108.  Second, we also compared the immunological response of vaccinated calves against pilus protein of M. bovis Epp-63 (300). If a ……

Linne 125. It seems that there is a large variation in  age on the primary dose inoculation. Any reason for this?

Line 128. Delete “all”

Line. Delete “The”

Line142….. …..672 (56.09%). The

Line 150. Kindly check custom or customized?

Line 174. Paired serum samples (with 2-3 weeks apart)  were obtained from….

Line 175-176.  Kindly delete “ The first serum…….dose.

Line 219-221. Kindly revisit whether it is necessary to mention this statement?

Line Figure 1. Sorry, I am confused with unit of Y-axis (may be %?)

Line 256 delete treatment group

Line Figure 2. I think there is no need to add post decimal digits in the Y-axis as author did in Figure 1. Sorry, I also could not find the description  of  */**  (at the foot notes ) embedded inside bars.

Line 261. 266, 267.. 273, 293,294, 319,358 Kindly delete “ value” in all document. Moreover, I could not understand what authors mean p< 0.001 while mentioning */**

Line 285 288. May be shifted to Ms &Ms section where authors describe the terms/phrases

Line289-291.  The figure may be omitted to save pages if the statement may be rephrased as “The retreatment was less required (P=0.2) in the autogenous group (21.4%) when compared to either commercial group (27.9%) or the sham group (34.4%). “

Line  312. …..age ate probably it was age at…..

Line Figure 7 unnecessary as the complete information has been mentioned in Line No 231-234. So it is redundancy.

Line Same is for Figure 8. However, authors may add SD/SE in the hard data while mentioning in the text to avoid Figure 8.

Line 355, 356, 362. Kindly follow the Units as for journal (lbs or Kg?)

Line 379. Sorry I could not find whether bulls has higher incidence in the Result Chapter.

Line 429. May we add a new subheading as “ Limitations”

Author Response

Authors planned a randomized controlled trail to investigate the comparison of two vaccines using different antigens from different pathogens and assessed their efficacy in terms of clinical outcomes of IBK. The trail was conducted for consecutive 5 years while data of only three years are presented due to low IBK incidence and/or outbreaks of pneumonia. The trail is well planned and objectives are very clear.

However, some minor changes suggested are as

Title may be suggestive as: A comparative study to assess the efficacy of an autogenous vaccine with a commercial vaccine against infectious bovine keratoconjunctivitis-A randomized controlled clinical trial.

We have adjusted the title with the suggestions in mind.

Line 19. ………administered the respective vaccines approximately 21 days apart……..

Line 19 has been changed.

Line 25,26. Kindly add p-values in the parentheses while describing the hard data e.g.,  (24.5% vs 30.06% vs 30.3% respectively: P?).

P values have been added to the abstract.

Line26,27. Same may be for Ig response and correlations with IBK.

P values have been added to this sentence.

Line45. The IBK has significant economic impact…..

The sentence has been edited.

Line 54-58. Various bacteria and viruses have also been found to be associated with IBK outbreaks, especially Mycoplasma bovoculi and bovine herpesvirus-1 that are  inducing ocular disease, but with clinical signs slightly different from those most commonly associated with IBK (12-14).

These sentences have been changed.

Line 101-105……to investigate the potential differences in efficacy between autogenous vaccine and a commercial vaccine in calves on IBK incidence (primary outcome) and IBK treatment success and adjusted weaning weight as secondary outcomes.

These lines have been changed.

Line 106-108.  Second, we also compared the immunological response of vaccinated calves against pilus protein of M. bovis Epp-63 (300). If a ……

The sentence has been changed.

Linne 125. It seems that there is a large variation in  age on the primary dose inoculation. Any reason for this?

We appreciate this comment. Since this was a field trial, many variables such as age and vaccination timing were challenging due to normal production practices that were subject to numerous factors.  This age range was for the entire study (5 years). The date of initial vaccination varied slightly from year to year due to weather/labor logistical issues. Additionally, the herd heavily utilized artificial insemination, so breeding dates varied slightly from year to year which would have affected birth dates.  The calving season each year had “outliers” with calves born early or late compared to the rest of the herd. As the years went by, the age range had the potential to only get wider. However, most of calves in this trial were born within a fairly slight window.

Line 128. Delete “all”

This entire sentence was deleted per Reviewer #1 recommendation.

Line. Delete “The”

No line was given for this suggestion, but I believe Line 132 was the most likely sentence. “The” has been removed ahead of the word data.

Line142….. …..672 (56.09%). The

The percentage of calves that were analyzed relative to the total study enrollment number has been added.

Line 150. Kindly check custom or customized?

The manufacturer has no “normal” autogenous formulation and all formulations are made with unique isolates sent in by the customer. Therefore, we believe custom is more grammatically correct here.

Line 174. Paired serum samples (with 2-3 weeks apart)  were obtained from….

This sentence has been edited. The serum samples were actually 4-5 weeks apart, since there were usually about two weeks between the first and second vaccination.

Line 175-176.  Kindly delete “ The first serum…….dose.

We believe the original sentence is needed to relay the time of serum collection relative to the vaccine administration.

Line 219-221. Kindly revisit whether it is necessary to mention this statement?

This sentence has been deleted.

Line Figure 1. Sorry, I am confused with unit of Y-axis (may be %?)

The Y axis is in individual IBK diagnoses. The Y axis has been changed to “Number of animals with IBK” for clairity

Line 256 delete treatment group

Treatment group has been deleted.

Line Figure 2. I think there is no need to add post decimal digits in the Y-axis as author did in Figure 1. Sorry, I also could not find the description  of  */**  (at the foot notes ) embedded inside bars.

The Y axis has been adjusted in the new figure. * and ** indicate significantly different values. A sentence has been added to each figure legend that uses these symbols to clarify the meaning.

Line 261. 266, 267.. 273, 293,294, 319,358 Kindly delete “ value” in all document. Moreover, I could not understand what authors mean p< 0.001 while mentioning */**

The word “value” has been deleted throughout the document when mentioning P. The clarification of */** from the previous reviewer suggestion should help clarify the correlation of these symbols with P values.

Line 285 288. May be shifted to Ms &Ms section where authors describe the terms/phrases

This portion has been moved to 2.1 of M & M.

Line289-291.  The figure may be omitted to save pages if the statement may be rephrased as “The retreatment was less required (P=0.2) in the autogenous group (21.4%) when compared to either commercial group (27.9%) or the sham group (34.4%). “

The sentence has been changed and the figure removed as suggested.

Line  312. …..age ate probably it was age at…..

Sentence has been edited.

Line Figure 7 unnecessary as the complete information has been mentioned in Line No 231-234. So it is redundancy.

Figure 7 has been removed.

Line Same is for Figure 8. However, authors may add SD/SE in the hard data while mentioning in the text to avoid Figure 8.

Figure 8 has been removed and the SD mentioned in the text.

Line 355, 356, 362. Kindly follow the Units as for journal (lbs or Kg?)

All values have been changed to kg units in 3.5.

Line 379. Sorry I could not find whether bulls has higher incidence in the Result Chapter.

The increased incidence of IBK in bull calves is mentioned on line 264-265 of the original submission (approximately 283 in the edited version). This is the second sentence in the paragraph after figure 2.

Line 429. May we add a new subheading as “ Limitations”

A subheading has been placed at this location.

Please see the attached changes tracked, edited manuscript file to view the detailed edits made to address the comments.  

Round 2

Reviewer 1 Report

The authors have revised this manuscript  as required.